# Study of Cytotoxicity and Internalization of Redox-Responsive Iron Oxide Nanoparticles on PC-3 and 4T1 Cancer Cell Lines

**DOI:** 10.3390/pharmaceutics15010127

**Published:** 2022-12-30

**Authors:** Timur R. Nizamov, Artem R. Iliasov, Stepan S. Vodopyanov, Irina V. Kozhina, Igor G. Bordyuzhin, Dmitry G. Zhukov, Anna V. Ivanova, Elizaveta S. Permyakova, Pavel S. Mogilnikov, Daniil A. Vishnevskiy, Igor V. Shchetinin, Maxim A. Abakumov, Alexander G. Savchenko

**Affiliations:** 1Department of Physical Materials Science, National University of Science and Technology (MISIS), Moscow 119049, Russia; 2Department of Invertebrate Zoology, Faculty of Biology, Lomonosov Moscow State University, Moscow 119234, Russia; 3Department of Medical Nanobiotechnology, N.I. Pirogov Russian National Research Medical University, Moscow 117997, Russia

**Keywords:** magnetic nanoparticles, redox-responsive materials, disulfides, cytotoxicity, tumor cells

## Abstract

Redox-responsive and magnetic nanomaterials are widely used in tumor treatment separately, and while the application of their combined functionalities is perspective, exactly how such synergistic effects can be implemented is still unclear. This report investigates the internalization dynamics of magnetic redox-responsive nanoparticles (MNP-SS) and their cytotoxicity toward PC-3 and 4T1 cell lines. It is shown that MNP-SS synthesized by covalent grafting of polyethylene glycol (PEG) on the magnetic nanoparticle (MNP) surface via SS-bonds lose their colloidal stability and aggregate fully in a solution containing DTT, and partially in conditioned media, whereas the PEGylated MNP (MNP-PEG) without S-S linker control remains stable under the same conditions. Internalized MNP-SS lose the PEG shell more quickly, causing enhanced magnetic core dissolution and thus increased toxicity. This was confirmed by fluorescence microscopy using MNP-SS dual-labeled by Cy3 via labile disulfide, and Cy5 via a rigid linker. The dyes demonstrated a significant difference in fluorescence dynamics and intensity. Additionally, MNP-SS demonstrate quicker cellular uptake compared to MNP-PEG, as confirmed by TEM analysis. The combination of disulfide bonds, leading to faster dissolution of the iron oxide core, and the high-oxidative potential Fe^3+^ ions can synergically enhance oxidative stress in comparison with more stable coating without SS-bonds in the case of MNP-PEG. It decreases the cancer cell viability, especially for the 4T1, which is known for being sensitive to ferroptosis-triggering factors. In this work, we have shown the effect of redox-responsive grafting of the MNP surface as a key factor affecting MNP-internalization rate and dissolution with the release of iron ions inside cancer cells. This kind of synergistic effect is described for the first time and can be used not only in combination with drug delivery, but also in treatment of tumors responsive to ferroptosis.

## 1. Introduction

In recent decades, the range of functional materials for biomedicine has been significantly expanded. These materials, with their unique nanoscale performance and special functional properties, have been used to formulate various drug-delivery systems, such as micelles, magnetic nanoparticles (MNP), polymersomes, and polymer complexes, [1]. Among these functional materials, stimulus-responsive drug-delivery systems are of great interest, due not only to their response to intracellular stimuli, but also their fast and efficient release of drugs in tumor cells [2,3]. These systems exhibit higher chemical stability in the bloodstream as well as a faster response to intracellular conditions, thus inducing drug release in the cytosol and cell nucleus, where most anticancer drugs exert their therapeutic effects. They usually react to specific internal stimuli such as pH [4], enzymes [5], and redox potential [6], triggering the drug release. Among them, the redox-responsive systems attract particular attention because of their wide and diverse applicability [7]. The high reduction potential in cells is explained mainly by glutathione (GSH), a reducing agent widely distributed in living cells [8,9]. It is known that the intracellular concentration of GSH is approximately 2–10 mM, especially in certain organelles such as the cytosol, mitochondria and cell nucleus, while the level of GSH in the extracellular environment (blood and extracellular matrix) is a thousand times lower (approximately 2–20 μM) [10]. In addition, the concentration of GSH in tumor tissues can be more than four times higher than in normal tissues, and can reach concentrations of the order of 100 mM [10]. It should be noted that endosomes and lysosomes also have a high reduction potential, which is controlled by the activity of induced γ-interferon-induced lysosomal thiol reductase (GILT) in the presence of L-cysteine [11,12,13,14]. This significant difference in properties between extracellular and intracellular environments, as well as between tumors and normal tissues, gives redox-responsive materials a unique advantage since they are stable in the extracellular environment, but quickly and efficiently release the drug inside the cell, which is the prerequisite for creating a broad class of diverse redox-responsive drug delivery vehicles with high selectivity and efficiency in antitumor therapy [7].

The high reducing potential of the intracellular environment is a trigger for redox-responsive systems, which contributes to the rapid and efficient release of the drug in tumor cells. Typically, these delivery vehicles are synthesized from redox-responsive compounds by incorporating a labile disulfide bond into their structure. The synthesis of redox-responsive polymers has been carried out using disulfide bonds: cystamine [15], 3,3′-dithiodipropionic acid [3], pyridyl disulfide methyl acrylate [16,17], dihydroxyethyl disulfide [18] and other obtained disulfide bonds based on lipoic acid [19], cysteine [20]. Their in vitro and in vivo drug release efficiency and antitumor efficacy have also been shown.

In addition to the above, it is worth noting that, currently, redox-responsive systems based on disulfide compounds are assembled from hydrophobic blocks of polycaprolactone [21,22], polylactic acid [17] and polylactic-co-glycolic acid polymers [23], and hydrophilic blocks of polyethylene glycol (PEG) [3,4,17,18,24], polyacrylic acid [25] and dextran [22]. PEG is the most used hydrophilic polymer. For example, the authors developed biodegradable micelles with PEG shells for rapid intracellular release of doxorubicin, and using micelles designed with a disulfide-binder diblock copolymer [21]. Despite the significant advantages of PEGylated compounds, the PEG coating adversely limits the rapid release of the drug from the delivery system due to its diffusion barrier, which can limit the use of PEG [24]. The usage of disulfide-based polymers can help to overcome this disadvantage as well.

However, the functional properties of these polymeric materials are often limited by the response. Furthermore, the use of MNP in the design of the above-described redox systems [16,21,22] is not so well-described in literature, although the methods for preparing, and subsequently functionalizing, magnetic nanomaterials are relatively well studied, and their use could significantly expand the scope of redox-responsive materials in biomedicine. This is because MNP are actively used in MRI diagnostics, magnetic hyperthermia and other areas of biomedicine due to their ability to be controlled remotely by applying an external magnetic field.

On the other hand, nanomaterials based on iron oxide can significantly affect the redox environment of the cell. Iron (III) ions have a redox potential of E (Fe^3+^/Fe^2+^) = +0.77 V [26], causing significant oxidative stress and starting a process called ferroptosis, a specific kind of Fe (III)-dependent programmed cell death, which is specified by the accumulation of lipids and phospholipid peroxides [27,28]. This process is biochemically different from other pathways of programmed cell death such as apoptosis. It is initiated by the malfunction of the GSH-dependent antioxidant system, leading to lipid peroxidation which, in turn, triggers subsequent programmed cell death. Nowadays, ferroptosis attracts particular attention for its potential use in anticancer therapy as this process may be triggered selectively in targeted cancer cells. Since some of the molecules controlling ferroptosis play important roles in metabolic pathways regulating cysteine exploitation, lipid peroxidation, iron and GSH homeostasis, there may be plenty of targets to disrupt and subsequently cause cell death [29,30]. Moreover, some specific kinds of tumors are more sensitive to the factors triggering ferroptosis; for example, several breast cancer cell lines, such as 4T1 [29]. The induction of ferroptosis can help to overcome tumor drug resistance [31,32,33,34].

The combination of iron oxide nanoparticles with redox-responsive functionality can be considered not only for targeted drug-delivery, but also provide a promising synergetic multifunctional nanomaterial for cancer treatment that can simultaneously affect iron and GSH homeostasis. Here, we report a study of cytotoxicity and internalization dynamics of redox-responsive iron oxide MNP on the PC-3 and 4T1 cancer cell lines. The structure, magnetic properties, size and morphology of the initial MNP were studied. Then the MNP were PEGylated using 2-nitrodopamine (NDP) as a linker. In the case of redox-responsive nanoparticles, an extra disulfide fragment was added as well. The colloidal stability was studied in reducing and conditioned media. The internalization dynamics were studied by fluorescence microscopy using Cy3 and Cy5 dual-labeled redox-responsive nanoparticles and by TEM study over time.

## 2. Materials and Methods

### 2.1. Materials

The following reagents were manufactured by Sigma Aldrich, USA: Iron(III) acetylacetonate (≥97.0%), Triethylene glycol (TEG) (99%), Sodium nitrite (NaNO_2_) (≥97.0%), Sodium hydrocarbonate (NaHCO_3_) (≥99.7%), Sulfuric acid (H_2_SO_4_) (95–98%), Phosphate-buffered saline (PBS), Dopamine hydrochloride (MQ 200), 3,3′-dithiodipropionic acid NHS ester (NHS-DTDP), N-Hydroxysuccinimide (NHS) (≥ 97.0%), Polyethylene glycol-acetic acid 2-aminoethyl ester, M = 3500 g/mol (NH_2_-PEG-COOH), α,ω -bis{2-[(3-carboxy-1-hydroxypropyl)amino]ethyl}polyethylene glycol, M = 3000 g/mol (PEG-COOH), N,N-Dimethylformamide (DMF) (≥99.8%), 1-Ethyl-3-(3-dimethylaminopropyl)carbodiimide (EDC) (≥97.0%), N,N-Diisopropylethylamine (DIPEA) (≥99%), Dithiothreitol (DTT) (≥95%), 3,3′-dithiodipropionic acid (DTDP) (≥99%), Dimethyl sulfoxide (DMSO) (≥99%), and Epon resin. 3-(4,5-dimethylthiazol-2-yl)-2,5-diphenyl-2H-tetrazolium bromide (MTS) was manufactured by Promega, USA. Paraformaldehyde OH(CH_2_O)nH (n = 8–100) was manufactured by Biovitrum, Russia. Cy5 NHS-ether and Cy3 amine dyes were manufactured by Lumiprobe, Russia. Ethanol (95%) was manufactured by Komponent-reactive, Russia. Deionized water was obtained in the Laboratory of Biomedical nanomaterials, NUST MISIS, Russia.

### 2.2. Cell Lines

PC-3 human prostate cancer cells and 4T1 mouse breast cancer cells (American Type Culture Collection, Manassas, VA, USA) were cultured in a 5% CO_2_ atmosphere at 37 °C in DMEM/F12 medium containing 10% fetal bovine serum (Sigma-Aldrich, Burlington, VT, USA), 1% L-glutamine (Gibco, Waltham, MA, USA), and 1% antibiotics (penicillin and streptomycin).

### 2.3. Synthesis of MNP

MNP were obtained by thermal decomposition of a precursor iron (III) acetylacetonate in a TEG medium, according to the previously published procedure [35]. The reaction mixture consisted of 4 mmol (1.413 g) of iron (III) acetylacetonate and 40 mL of TEG solvent. After dehydration for 1 h at 120 °C, the reaction mixture was heated to 285 °C (TEG boiling point) at a rate of 3 °C per minute. After being heated to the required temperature, the temperature was maintained at 285 °C with magnetic stirring for 1 h. The mixture was then cooled naturally to room temperature (Figure 1A).

### 2.4. Synthesis of NDP

The nitration of dopamine was carried out using a previously published protocol [36]. A 0.500 g (2.63 mmol) amount of dopamine hydrochloride was dissolved in 15 mL of degassed water and 0.545 g (7.9 mmol) of NaNO_2_ was added with cooling. Then 2 mL of 1% H_2_SO_4_ (20 mg) was slowly added dropwise and stirred for 30 min at 0 °C. An aqueous solution containing 0.035 g of NaHCO_3_ was added to the reaction mixture and stirred for another 20 min. The orange precipitate was filtered off using a Schott filter and washed with ice water. The obtained NDP substance was dried on a rotary evaporator.

### 2.5. Synthesis of Redox-Responsive MNP (MNP-SS)

To obtain MNP-SS, 318.5 mg of NH_2_-PEG-COOH was dissolved in 2 mL of DMF and 30.7 mg of NHS-DTDP in 1 mL of DMF. Then, 200 µL of the NH2-PEG-COOH solution was mixed with 100 µL of the NHS-DTDP solution, and 200 µL of the DMF solution with 0.9 mg of NDP was added to the resulting solution. The resulting solution was left to stir for 2 h. Next, 450 µL of the solution was mixed with 500 µL of the MNP solution in TEG at a concentration of 7.73 mg/mL and left to stir on a shaker for a day. At this stage, the resulting sample was either washed from the organic solvent or used to introduce a fluorescent dye. In the first case, the sample was diluted with 2 mL of water and dialyzed against 200 mL of water in a dialysis tube with a pore size of 12 kDa for 2 days. The water was changed twice. The scheme of the MNP-SS synthesis and the MNP-SS proposed structure are shown in Figure 1A and C, respectively.

### 2.6. Synthesis of MNP-SS with Cy3 and Cy5 Fluorescent Dyes

An amount of 7.5 mg Cy5 NHS-ester dye was dissolved in 750 μL DMF. Next, 4 mg of NDP was added and the solution was left to mix on a shaker for 2 h. Next, 10 µL of the Cy5 conjugate with NDP solution was injected (before dialysis tubing step) into 950 µL of the MNP-SS obtained at the previous stage and left to mix on a shaker for a day. After that, the sample was diluted with 2 mL of water and placed on dialysis against 200 mL of water in a dialysis tube with a pore size of 12 kDa for 2 days. The water was changed twice.

After the dialysis, the iron concentration was measured using a ferrozine test and the sample was concentrated to a concentration of 4 mg/mL by centrifugal filtration. 1 mL of the sample was diluted with an equal volume of 2× PBS, after which 5.7 µL of 1% NHS solution in DMF and 9.6 µL of 1% EDC solution in DMF were added to the solution. Half an hour later, 7 μL of 1% Cy3–NH2 solution in DMF was added to the solution and left to stir overnight. Next, the sample was placed on dialysis against water with a volume of 200 mL in a dialysis bag with a pore size of 12 kDa for 2 days. The water was changed twice. The resulting solution was eluted twice through a Sephadex column to get rid of traces of free dyes. The final structure of the dual-labeled MNP-SS with Cy5 and Cy3 is presented in Figure 1B.

### 2.7. Preparation of Control MNP-PEG

To evaluate the effectiveness of MNP-SS, the same sample, but without an S-S linker, was synthesized as a reference. An amount of 50 mg of PEG-COOH was dissolved in 0.5 mL of DMF, after which 191 μL of 1% NHS and 320 μL of 1% EDC solution in DMF were added to the resulting solution and left for 30 min with active stirring on a shaker. Next, 450 µL of the solution of the resulting NHS ester was mixed with 300 µL of a 0.5% solution of NDP in DMF (5 mg/mL) and 7 µL of DIPEA was added. After complete dissolution of NDP, the solution was mixed with 0.75 mL of a solution of MNP in TEG and left to mix on a shaker for a day. After that, the sample was diluted with 2 mL of water and placed on dialysis against 200 mL of water in a dialysis tubing with a pore size of 12 kDa for 2 days. The water was changed twice. A condensed diagram of MNP-PEG synthesis is given in Figure 1A.

### 2.8. Study of the Reductant Effect on Disulfide Bond-Dissociation Dynamics

To study the behavior of redox-responsive MNP in a reducing medium, the following experiments were carried out in an aqueous and PBS medium with pH = 7.4 with the addition of the reducing agent DTT at different concentrations (12.5; 25; 50; 100 mM).

First, 0.05 mL of redox-responsive MNP in an aqueous medium with C = 2 mg/mL was diluted in water and a subsequent amount of 0.5 M DTT solution was added to reach the indicated concentrations. The final volume of the solutions was 1 mL. The hydrodynamic size measurements of the obtained four samples were carried out 0, 30 and 60 min after the mixing.

Then, a comparison of the colloidal stability of control and redox-responsive MNP under the conditions simulating the cellular media was carried out in a DTT solution with a concentration of 25 mM in PBS with pH = 7.4. The as-prepared 1 mL of simulating solution was mixed with either 0.05 mL of redox-responsive or control MNP solutions with C = 2 mg/mL. The hydrodynamic size of the resulting solution was measured 0, 30, 60 and 90 min after mixing.

The effect of the reducing agent concentration under conditions simulating a cellular environment on the colloidal stability of redox-responsive MNP was studied as well. A series of DTT-aqueous solutions with concentrations of 12.5, 25, 50, and 100 mM in PBS with pH = 7.4 were prepared for this reason. Then, 0.05 mL of redox-responsive nanoparticles were mixed with 1 mL of as-prepared buffers with DTT at the indicated concentrations. The hydrodynamic sizes of the obtained four samples were measured 0, 30, 60, and 90 min after mixing.

### 2.9. Stability of MNP-PEG and MNP-SS in Conditioned Media

To study stability, a conditioned medium DMEM F12 with 10% FBS was used in which PC-3 cells had grown during 2–3 division cycles (48–72 h). The hydrodynamic diameter was measured by dynamic light scattering (DLS) using a Malvern Zetasizer Nano ZEN3600 (Malvern Instruments Ltd., Malvern, UK). The samples were diluted to a final Fe^3+^ concentration of 0.4 mg/mL with conditioned media DMEM F12 and were measured at a temperature of 25°C. Z-average size was used for all nanoparticles diameter determination. Measurements were carried out after 30, 60, 120 and 180 min (storage of nanoparticles at +37 °C in the DMEM F12-conditioned media).

### 2.10. Study of Internalization Dynamics and Colocalization

PC-3 cells (3 × 10^5^ cells) were seeded in a 30-mm SPL cover glass dish (Biolab, Seoul, Korea) and, after 24 h, were treated with Cy3, Cy5-labeled MNP-SS (final concentration 100 µg/mL Fe_2_O_3_). Then cells were fixed with 4% formalin solution at the following time points: 1 h, 4 h, 24 h, 48 h, 72 h. The experiments were carried out in triplicate. The study of internalization process by living cells, analysis of accumulation and colocalization were carried out using a confocal multiphoton microscope Nikon A1R MP (Nikon, Japan; oil, immersion objective ×60/1.49).

### 2.11. Cytotoxicity Study

Cell cultures PC-3 and 4T1 were seeded in 96-well plates at 10^4^ and 12 × 10^3^ cells per well, respectively. After 24 h, nanoparticles were added to them for incubation for 48 h. After that, the MTS dye was added to the wells for 4 h, after which the medium optical density was measured at a wavelength of 490 nm. The obtained values for the wells in which the nanoparticles were added were compared with the optical density of control wells.

### 2.12. TEM Study of MNP Uptake and Distribution in Cells

The capture dynamics of MNP-PEG and MNP-SS nanoparticles were studied in vitro on PC-3 (human prostate adenocarcinoma) and 4T1 (mouse breast carcinoma) tumor cells using transmission electron microscopy. The cells were seeded in the wells of a special chamber with a glass bottom (Ibidi), incubated in a growth medium according to a standard protocol until 50% confluence of the monolayer was reached, then the medium was replaced with medium with nanoparticles at a concentration of 200 ug/mL and incubated for 2 h or 24 h. After incubation with nanoparticles, the cells were washed from the medium with nanoparticles using DPBS++ phosphate buffer and fixed (2.5% glutaraldehyde, 0.5% formaldehyde in DPBS++ buffer for 30 min, RT), then washed in DPBS++ and post-fixed (1% OsO_4_ on DPBS++ for 1 h, RT). Next, the fixed cells were dehydrated through ascending ethanol series according to the standard protocol and finally the cells were embedded in Epon resin. Ultrathin 70-nm-thick sections of cells were made on the Leica EM UC-6 (Leica) ultratome. The sections were contrasted with Uranyless and lead citrate and studied using transmission electron microscopy.

### 2.13. Statistics

Tests for cytotoxicity were performed 3 times in triplicate. The data presented in the histograms contain the mean value of cell viability ± SD (standard deviation). The statistical significance of differences between the groups was determined using a parametric paired one-tailed *t*-test. Differences were considered statistically significant at * *p* < 0.05, ** *p* < 0.01, *** *p* < 0.001.

### 2.14. Methods of Characterization

Transmission electron microscopy (TEM). JEOL JEM-1400 microscope (JEOL Ltd.; Tokyo, Japan) with an accelerating voltage of 120 kV was used to study the synthesized MNP and ultrathin cell sections.

X-ray diffraction analysis (XRD). The structural phase analysis was studied on a Rigaku Ultima IV diffractometer (Rigaku; Tokyo, Japan) using CuKα radiation and a graphite monochromator.

Magnetic measurements (VSM). The hysteresis properties of the powders were measured on a VSM-250 vibrating magnetometer (YP Magnetic Technology; Changchun, China) in fields up to 2 T at room temperature.

Dynamic light scattering (DLS). The hydrodynamic diameter of MNP and their zeta potential were measured using Zetasizer Nano ZS equipment (Malvern; Kassel, Germany). The volume of the measured NP solution varied from 1 to 2 mL. Every sample was measured 3 times.

Thermal gravimetric analysis (TGA). TGA was performed using the simultaneous thermal analyzer Netzsch STA 449 F3 (NETZSCH; Selb, Germany). The samples were placed into alundum crucibles and heated in the temperature range from 50 to 800 °C at 10 °C/min under argon flow. Before the analysis, solvents were removed from the samples by evaporation using a rotary evaporator.

Measurement of iron concentration. A series of solutions were prepared with concentrations of 0.1, 0.25, 0.5, 0.75, 1.0, 1.5, 2 mg/mL from the ICP standard for iron. Then, 100 µL of a sample with an unknown iron concentration was dissolved in 400 µL of concentrated hydrochloric acid for 2 h. Next, the solution was diluted 100 times with deionized water, and 400 µL of the resulting solution was mixed with 200 µL of deionized water and 40 µL of the ferrozine test. After 5 min, 300 μL of the resulting solution was placed into two wells of a 96-well plate and absorbance was measured at λ = 560 nm on a Thermo Scientific Multiskan GO spectrophotometer (Thermo Fisher Scientific Corporation; Waltham, MA, USA) in the photometry mode. According to the calibration curve, the concentration of iron was determined.

Fourier transform infrared spectroscopy (FT-IR). After the PEGylation and the dialysis tubing, the samples were dried on a rotary evaporator until powder formation. Then the powders were characterized via FT-IR using a Bruker Vertex 70v vacuum spectrometer (Bruker Corporation; Billerica, MA, USA) in the range of 400–4000 cm^−1^.

## 3. Results and Discussion

### 3.1. Characterization of the Synthesized MNP

The MNP were obtained by thermal decomposition of the iron (III) acetylacetonate complex in glycol media. Through this method, monodisperse iron oxide particles of controlled size can be obtained [35,37]. Moreover, the medium used in this approach is non-aqueous polar TEG and no surfactant was added, thus a surface of the as-synthesized nanoparticles is only weakly stabilized by the glycol molecules. These factors significantly simplify the subsequent strategy of MNP-surface modification, because the surface modifier (PEG conjugate) can be added directly into the MNP solution without a preliminary step of nanoparticle washing and separation.

The synthesized MNP were studied by TEM to determine the size, shape and size distribution of the nanoparticles. The MNP have a nearly spherical shape and an average size of 6.9 ± 1.7 nm (Figure 2A). According to DLS measurements, the average solvodynamic size is 8.9 ± 1.6 nm (Figure 2B), which is a bit more than the average size by TEM as expected, because DLS measures the particle core with solvatic shell of adsorbed stabilizer. The XRD studies revealed that the MNP have an inverse spinel structure, specific for magnetite or maghemite or even a mixture of them (Figure 2C). These two magnetic iron oxides have a similar structure, and both show relatively similar magnetic properties. As the measured lattice parameter a = 8.388 Å is closer to magnetite (a = 8.396 Å) than to maghemite (a = 8.3515 Å), the obtained iron oxide phase can conditionally be described as magnetite [38]. The crystallite size is 6.4 ± 0.5 nm, which is slightly less than the average size by TEM and DLS, which means that the synthesized MNP have a monocrystalline structure. According to magnetic measurements, the saturation magnetization σ_s_ of MNP is 68.8 ± 0.2 Am^2^/kg, remanence magnetization σ_r_ is 2.0 ± 0.1 Am^2^/kg, and coercive force H_c_ is 2 ± 2 kA/m in the magnetic field of 2 T (Figure 2D). The obtained magnetic parameters are specific for magnetite, a soft ferrimagnetic material. In contrast, the MNP average size is 7 nm, which is under their critical size of ~20 nm, and they should be superparamagnetic, where a non-zero H_c_ is observed [39]. This may result from a peculiarity of the magnetic properties-measuring equipment VSM-250, which is made for measurement of hard magnetic properties. In the case of soft magnetic or paramagnetic materials, it measures a Hc with high error at near zero fields.

### 3.2. PEGylation of MNP and Colloidal Stability of MNP-SS and MNP-PEG in Reducing Media

The MNP-PEG and MNP-SS were both modified by covalent grafting of the PEG chains via NDP, which forms very stable complexes. The chelating fragment of the catechol molecule, which consists of vicinal hydroxyls, strongly interacts with iron atoms on the MNP surface [40]. In the case of MNP-SS, an extra disulfide bond was also introduced into the surface modifier’s structure (Figure 1C). A disulfide, being a more labile bond, can be easily cleaved by reductive factors of the media, which leads to desorption of the surface stabilizer and subsequent aggregation of nanoparticle cores [41,42]. In living cells, this process usually happens due to the presence of GSH, GILT and other reducing factors in the cytoplasm or organelles [10,43]. Therefore, the stability of the modified nanoparticles was first studied in solutions simulating reducing conditions. The PBS buffer with pH = 7.4 and DTT reducing agent of various concentrations was used for this purpose. DTT is widely used in biochemistry as a redox agent [44]. Its molecule has 2 thiol groups that easily reduce disulfide bonds of various organic compounds in media with pH above 7, where the redox potential increases to −0.33 V.

The MNP-PEG and MNP-SS were dispersed in a simulated media with C (DTT) = 25 mM. According to the DLS measurement of these solutions over time, the MNP-PEG remained stable for more than 900 min or 15 h with no significant hydrodynamic size change, while the MNP-SS started to aggregate after 60 min of treatment—the average hydrodynamic size increased from 50 nm to 68 nm (Figure 3A). After 90 min of treatment the average size drastically increased to 1500 nm with the formation of a visible precipitate. The reduction of disulfides led to the desorption of PEG conjugate from the nanoparticle surface, and only sulfhydryl groups remained. This process caused destabilization of the solution and subsequent aggregation of the MNP-SS sample (Figure 1B). The effect of DDT concentration on the dynamics of aggregation was also studied in a 12.5–100 mM diapason and a concentration dependence was observed (Figure 3B). It should be noted that neither such dependence nor aggregation was detected in deionized water without PBS buffer, where MNP-SS remained stable for the same period in the same concentration ranges of DTT (Figure 3C). Such a drastic difference in colloidal stability between distilled water and PBS buffer is probably due to sufficiently high ionic strength and pH in the PBS buffer. It is enough to cause the rapid reduction of disulfide bonds and desorption of PEG from the MNP surface with a subsequent loss of colloidal stability.

Cancer cells during the cell metabolism usually excrete various metabolites into the intracellular media. Among them are GSH and other products that may play the role of reducing agents [45,46]. Therefore, when redox-responsive nanoparticles are added to the cancer cell medium, the metabolites can cause some reduction processes in their disulfide sites, and even aggregation. To study this effect, the colloidal stability of MNP-PEG and MNP-SS was measured in conditioned medium at 37 °C after PC-3 cell cultivation (Figure 3D). The MNP-PEG sample remained stable and monodispersed throughout the study period—the average hydrodynamic size varied from 27 ± 1.1 nm to 28 ± 1.4 nm. Meanwhile, the MNP-SS became polydispersed, and the average hydrodynamic size increased and fluctuated from 51 ± 7 nm to 135 ± 24 nm, but no full precipitation was observed. It was supposed that traces of reducing agents remaining in conditioned medium may have caused the partial aggregation of MNP-SS, but were not enough to trigger total precipitation of the sample. This process led to an increase of average size and polydispersity. The MNP-PEG remained unchanged due to the absence of redox-responsive disulfide in its surface-stabilizing shell.

### 3.3. FT-IR Characterization of the MNP-PEG and MNP-SS Samples

MNP-PEG and MNP-SS were studied by FT-IR (Figure 4). Both spectra consist of several bands indicating the PEG presence: intensive bands at 1100 cm^−1^ and 2860 cm^−1^, specific for the C-O-C of aliphatic ethers and for the C-H alkane bands, respectively [47]. The 1720 cm^−1^ wavenumber can be attributed to the C=O of carboxylic acids or esters of the PEGs [48]. The NDP was detected at 1542 cm^−1^, 1488 cm^−1^, 1344 cm^−1^, 1323 cm^−1^, and 1278 cm^−1^ wavenumbers, corresponding with the N-O of the asymmetric nitro-compound; the C=C double bonds of the aromatic ring; the O-H band of catechol; the N-O of the symmetric nitro-compound and C-O phenolic stretching, respectively [40]. The covalent amide bond between PEG-COOH and NDP of the MNP-PEG was detected at the 1647 cm^−1^ (C=O amide bending) and 1242 cm^−1^ (C-N amide stretching) wavenumbers [49]. In the case of MNP-SS, these bands correspond with the amide band between DTDP and NH2-PEG-COOH or NDP. The band specific for disulfide bonds was detected at the 490 cm^−1^ wavenumber; as expected, it was relatively weak, as sulfur at the − oxidative state usually has a weak appearance on FTIR spectra [50]. The 624 cm^−1^ and 549 cm^−1^ wavenumbers, specific for the Fe-O-bond, were present on the FTIR spectra as well, as expected for the PEG-modified MNP [47,51].

The obtained FT-IR data confirmed that the MNP-PEG and MNP-SS consist of a magnetic core covalently modified with PEG via NDP (and DTDP in the case of MNP-SS), as expected.

### 3.4. Study of Internalization Dynamics and Colocalization

The colocalization of Cy3 and Cy5-fluorescent labels covalently grafted to MNP-SS in a PC-3 cell culture was measured at different time points (Figure 5). The two labels were grafted differently to the MNP-SS surface. Initially, the NHS-ester of Cy5 was conjugated with NDP. In the next step, this conjugate was mixed with MNP-SS nanoparticles. The Cy5–NDP conjugate is smaller than the PEG–DTDP–NDP shell of MNP-SS, so it can easily diffuse through it and graft onto the magnetic core surface. Then, the Cy3-NH_2_ was covalently grafted to the PEG COOH-end via formation of an amide-bond. Finally, the Cy3 was attached to the nanoparticles via a labile SS-bond, while the Cy5 was attached rigidly. The general scheme of the dual-labelled MNP-SS is given above (Figure 1B).

The graph shows that the intensity of the Cy3-fluorescent label fades over time, while the intensity of Cy5-label, on the contrary, flares up (Figure 5A). We attribute these processes to the fact that nanoparticles, after entering the cell, degrade in the endosomal environment, being affected by low pH and high reducing activity. First, they lose the PEG shell, grafted via a SS-bond to the magnetic core, as well as the Cy3 label attached to the PEG. The Cy3 fluorescence can be seen even in the cell nucleus (Figure 5B). Since iron oxide is biodegradable, the remaining magnetic core gradually dissolves and loses the Cy5 label that had been rigidly grafted to its surface via NDP. As the PEG–Cy3 conjugate is being excreted by exocytosis, a decrease in the fluorescence intensity of Cy3 in the cytoplasm is observed (Figure 5C). As for Cy5, its initial intensity is quenched by the magnetic core, caused by an effect of Förster resonance energy transfer (FRET effect). This is a phenomenon of energy transfer from an excited donor molecule (fluorescent label) to an acceptor when the acceptor is located close to a donor, usually less than 5 nm [52]. This effect is also described for iron oxide nanoparticles and fluorescent labels grafted onto its surface [53]. Therefore, the Cy5 label, initially grafted onto the nanoparticles surface via short NDP linker, was quenched, while Cy3, grafted via a PEG–DTDP–NDP conjugate, which is much longer than NDP, remained fluorescent. When the magnetic core starts to dissolve in endosomes, it leads to desorption of Cy5 and the subsequent increase of fluorescence intensity.

At the first time points, the initial intensive fluorescence of Cy3 fades and becomes barely visible after 72 h due to excretion of the label, which is detached from the nanoparticles after the reduction of disulfide bonds. Later, the Cy5, which is initially rigidly grafted and quenched by the magnetic core, starts to flare up over time with the slow subsequent dissolution of the iron oxide surface in the acidic endosomal medium. Photographs of cell fluorescence for all studied periods of time with dual-labeled nanoparticles can be found in the Appendix A).

### 3.5. Assessment of Cytotoxicity

The cytotoxicity of MNP-PEG and MNP-SS was studied in PC-3 and 4T1 tumor cell cultures. Results are presented in Figure 6.

Initially, the cytotoxicity of MNP-PEG and MNP-SS was studied in the PC-3 human adenocarcinoma cell line. The MNP-PEG sample with no disulfide bonds showed no significant cytotoxicity in the concentration range tested, while redox-responsive MNP-SS led to a slight decrease in cell viability to ~90% at 600 µg/mL. Compared to PC-3, the 4T1 murine mammary carcinoma cell line was less viable (Figure 6B). This cell line, as well as other kinds of breast cancer stem cells, is very sensitive to ferroptosis-triggering factors [29,54]. Both types of nanoparticles were more toxic toward 4T1 cells, especially MNP-SS, as the 4T1 cell viability decreased to ~70% and ~30% at 600 µg/mL for the MNP-PEG and the MNP-SS, respectively. This may be due to high oxidative activity of the iron oxide core—a significant factor triggering ferroptosis [27]. The redox-responsive disulfide bonds of MNP-SS in combination with iron ions enhance the oxidative stress, and finally decrease the cell viability compared to control MNP-PEG. These two factors synergistically deplete the level of GSH in the cells which may trigger cell death.

The cytotoxicity of pristine DTDP, the disulfide compound introduced into the PEG shell structure of MNP-SS, was also evaluated in the PC-3 and 4T1 cell lines (Appendix A). No decrease in cell viability was observed at the investigated concentration range up to 250 µM. The chosen range approximately corresponds to the DTDP concentration in MNP-SS previously tested. The estimation of DTDP content in the MNP-SS was carried out by a TGA analysis (Appendix A). The sample loses ~25% mass of its organic shell, consisting of PEG, DTDP and NDP, at the 300–400 °C range. Thus, at 600 µg/mL of MNP-SS, the concentration of DTDP should be ~50 µM, which is in the investigated pristine-DTDP concentration range. Therefore, the increased MNP-SS cytotoxicity was caused by the synergetic combination of these oxidative factors of iron-containing magnetic core and disulfide shell. The same synergistic combination of Cu and Fe ions, causing cupperptosis/ferroptosis, significantly decreases breast cancer cells viability and is described for metalloorganic frameworks [55].

It was hypothesized that the decrease in the viability of 4T1 cells was caused by the synergetic effect of disulfide bonds of the MNP-SS polymeric shell together with the highly oxidative Fe (III) of the MNP magnetic core. They both cause the depletion of GSH stores in cells, which can lead to ferroptosis [30]. Therefore, the 4T1 cell line, which is sensitive to ferroptotic factors, showed significant decrease in viability after incubation with MNP-SS. The PC-3 cell, being less sensitive, demonstrated a smaller decrease in viability.

### 3.6. TEM Study of MNP Uptake and Distribution in Cells

For a more detailed look at what happens to nanoparticles inside cells during incubation, TEM imaging was performed over time. Two time points were chosen for that: 2 h and 24 h.

MNP-SS after 2 h incubation with PC-3 cells were found on the surface of the outer cell membrane and inside the cytoplasm (Figure 7A–D). The process of nanoparticle capture by the vesicles was observed. MNP-SS nanoparticles have electron-contrast cores with a darkened shell on the studied slices (Figure 7D and Figure 8D). An adsorption of heavy metals (Os, Pb, etc.), used for contrasting cell structures, on the MNP-SS led to the formation of the darkened shell. This is due to complexation of the metal ions with sulfide and hydrosulfide bonds on the nanoparticle surface. The nanoparticles were gathered in incompact clumps and easily found on the plasmalemma as well as inside the vesicles (Figure 7C). The observed vesicles with the nanoparticles after 2 h of incubation might be endosomes [56]. MNP-PEG nanoparticles after 2 h incubation with PC-3 cells were hardly found on TEM images of the studied samples, and only scarce single nanoparticles were found in vesicles (Figure 7E–H).

The same results were obtained with 4T1 cells: after 2 h of incubation, MNP-SS nanoparticles were easily found on plasmalemma as well as in vesicles (Figure 8A–D), but MNP-PEG were scarce on the studied sections (Figure 8E–H). The MNP-SS nanoparticles were again surrounded by a shaded shell of contrasting heavy metals.

As was observed for the both 4T1 and PC-3 cell lines, the MNP-SS show quicker uptake dynamics compared to MNP-PEG over a 2 h incubation period. This might be due to partial aggregation of the MNP-SS in the intercellular medium containing cell metabolites, some of which can be relatively active reducing agents, such as GSH, cysteine, etc. [45,46]. Tests on the colloidal stability in the conditioned medium showed that the MNP-SS partially aggregates and becomes more polydispersed (Figure 3D). This factor may impact MNP-SS uptake dynamics, because the larger nanoparticles have faster cellular uptake. Moreover, the faster uptake can enhance the overall cytotoxicity of the MNP-SS, increasing the local intracellular concentration of nanoparticles, and, subsequently, Fe (III) and disulfides as well, which are important factors of oxidative cell stress.

MNP-SS and MNP-PEG after 24 h incubation with PC-3 cells were gathered in numerous vesicles distributed all around the cytoplasm (Figure 9). The average amount of uptaken nanoparticles visually increased many times compared to the 2 h incubation time. No clear differences were noted in the accumulation patterns between nanoparticles. In both cases the vesicles were filled with nanoparticles, while no nanoparticles were observed on the plasmalemma. The observed vesicles with the nanoparticles after 24 h of incubation might be lysosomes. In contrast with the previous incubation time, the MNP-SS nanoparticles had no shaded shell after 24 h of incubation. This could be due to etching on the magnetic core surface and the subsequent desorption of the disulfide-containing shell. The acidic environment near pH ~4.7 and high reducing activity of GSH have led to the intensive dissolution of the iron oxide core and the reduction of disulfide bonds.

We could not perform the same experiments with 4T1 cells with a 24 h incubation period because the cells lost most of their adhesion contact with the well bottom after 24 h, and they were also lost during fixation and the subsequent steps of washing, dehydration and embedding. This pertained to both types of nanoparticles studied. The high cytotoxicity of the nanoparticles at the concentrations chosen for incubation could have finally led to the detachment of 4T1 cells as described.

TEM results showed that MNP-SS were actively absorbed by cancer cells in vitro after 2 h of incubation, but MNP-PEG were quite scarce on, and inside, the cells. After 24 h of incubation, both types of nanoparticles were collected into large aggregates in vesicles inside the cytoplasm of PC-3 cells. MNP-SS have faster uptake dynamics compared to MNP-PEG, which is caused by the partial aggregation of MNP-SS in the intercellular reducing media. The faster uptake enhances the MNP-SS cytotoxicity as well.

### 3.7. Conclusions

In this study, we first investigated the internalization dynamics of magnetic redox-responsive and control iron oxide magnetic nanoparticles by TEM study of cellular uptake and distribution over time, and cytotoxicity toward the PC-3 and the 4T1 tumor cell lines by the MTS-test. The internalization dynamics of MNP-SS, dual-labeled with Cy3 via labile disulfide and rigidly with Cy5, was studied by confocal fluorescent imaging as well. MNP-SS were synthesized by covalent grafting of PEG onto the magnetic iron oxide nanoparticle’s surface via disulfide bonds, while the MNP-PEG were left without disulfide bonds. The inversed spinel structure of the initial MNP was revealed through XRD analysis, which is specific both for magnetite and maghemite. The MNP are superparamagnetic, according to the magnetic measurements. The MNP successful PEGylation and introduction of disulfide were confirmed by FT-IR. The MNP-SS aggregated completely in reducing media and partially in conditioned media, while the MNP-PEG remained stable. MNP-SS showed higher cytotoxicity compared to MNP-PEG on PC-3, and especially on the 4T1 cells. The difference in 4T1 viability rate was approximately 35% for 600 ug/mL of MNP-SS. This is most likely related to a more rapid cellular uptake of MNP-SS. It was confirmed by TEM study of MNP uptake and distribution in cells. After a 2 h period, we observed an abundance of MNP-SS inside cells, but a smaller amount of MNP-PEG. MNP-SS lose their PEG shell inside cells more rapidly than MNP-PEG, causing enhanced magnetic core dissolution, which finally leads to increased cytotoxicity. According to the confocal fluorescence study, Cy3, which was labilely grafted via PEG–SS to the MNP, tends to be released first due to a faster reduction of disulfide bonds, while Cy5, which is rigidly grafted via short NDP to the MNP, was released later when the magnetic core began to dissolve. The dual effect of Fe (III) and disulfides may synergistically enhance the oxidative stress of cancer cells and cause cell death, presumably via the ferroptosis pathway, which is reported here for the first time. The most significant decrease in cell viability was shown for the 4T1 cell line, which is sensitive to ferroptosis-triggering factors. Therefore, combining disulfide redox-responsivity with iron oxide nanoparticles produces a promising functional nanomaterial, not only in drug delivery, but also for the treatment of tumors sensitive to ferroptosis-triggering factors.

## Figures and Tables

**Figure 1 pharmaceutics-15-00127-f001:**
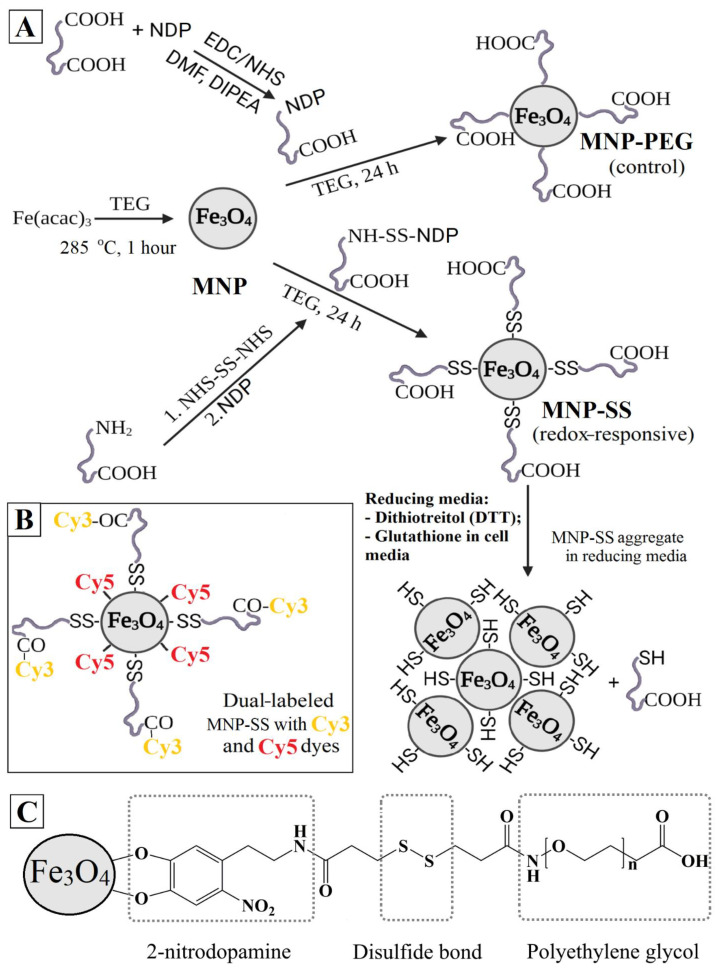
(**A**) Scheme of the MNP, MNP-PEG and MNP-SS synthetic pathway, and MNP-SS aggregation in a reducing medium; (**B**) Structure of the dual-labeled MNP-SS; (**C**) Structure of the MNP-SS.

**Figure 2 pharmaceutics-15-00127-f002:**
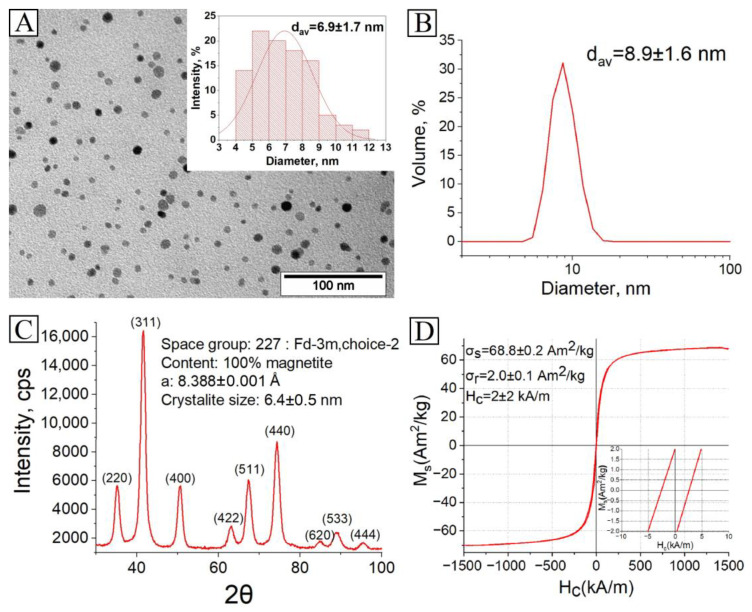
Characterization of the synthesized MNP: (**A**) TEM image and size distribution; (**B**) Size distribution by DLS of as-synthesized nanoparticles; (**C**) XRD spectrum; (**D**) Magnetic hysteresis loop.

**Figure 3 pharmaceutics-15-00127-f003:**
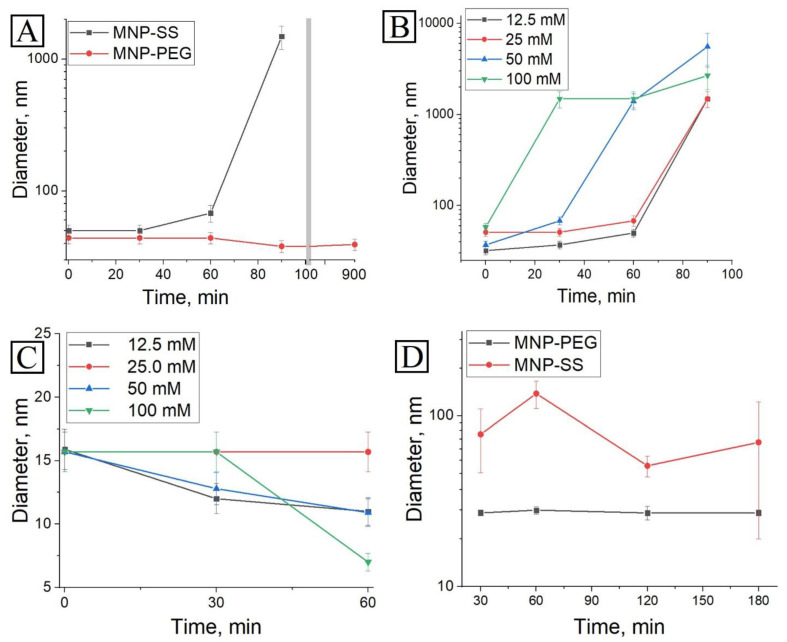
Measurement of hydrodynamic size dynamics in different conditions: (**A**) MNP-SS and MNP-PEG in PBS with pH = 7.4 and C (DTT) = 25 m; (**B**) MNP-SS in PBS with pH = 7.4 and C (DTT) = 12.5, 25, 50, or 100 mM; (**C**) MNP-SS in aqueous solutions of DTT with C (DTT) = 12.5, 25, 50, or 100 mM; (**A**,**D**) MNP-SS and MNP-PEG in conditioned medium after PC-3 cell culture growth.

**Figure 4 pharmaceutics-15-00127-f004:**
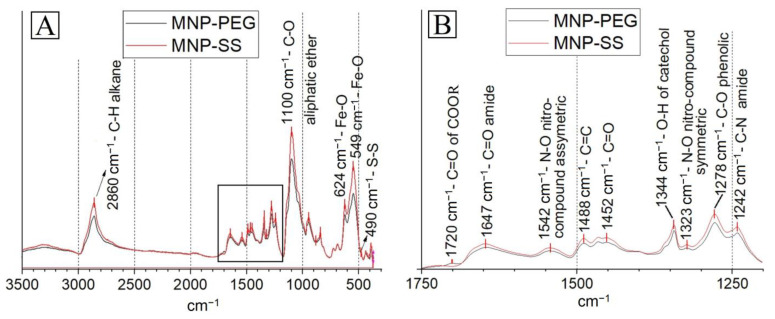
FT-IR spectra of the MNP-PEG and MNP-SS samples (**A**) and the same spectra at the 1750–1200 cm^−1^ range (**B**).

**Figure 5 pharmaceutics-15-00127-f005:**
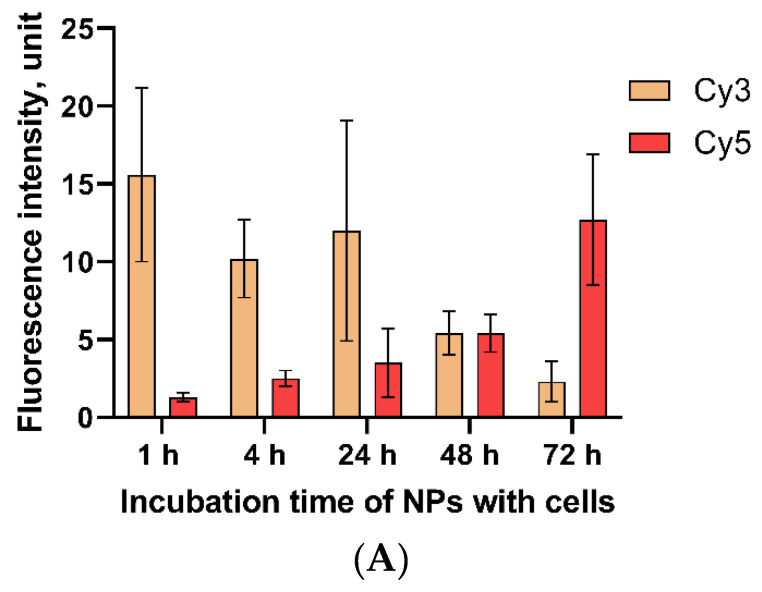
Changes in fluorescence intensity of Cy3 and Cy5 dyes attached to MNP in cells (**A**) and representative images of cells incubated with Cy3–Cy5-labeled NPs at the first time point at 1 h (**B**) and last time point at 72 h (**C**). The bar graph contains the mean value ± SD.

**Figure 6 pharmaceutics-15-00127-f006:**
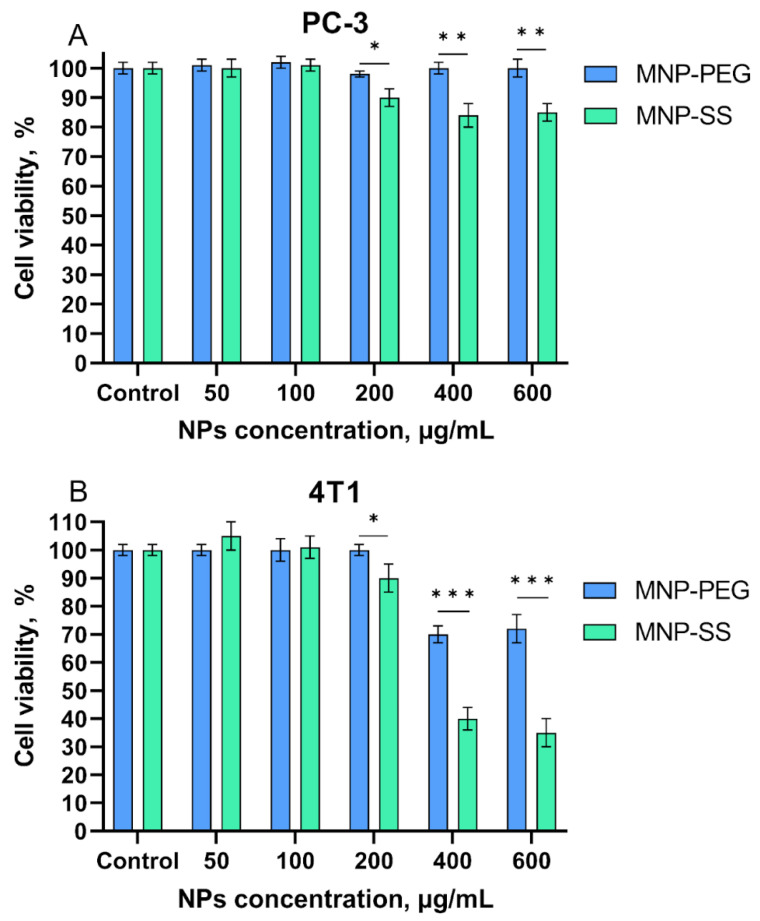
Evaluation of cytotoxicity of MNP-PEG and MNP-SS on PC-3 (**A**) and 4T1 (**B**) cells. Differences were considered statistically significant at: *—*p* < 0.05, **—*p* < 0.01, ***—*p* < 0.001.

**Figure 7 pharmaceutics-15-00127-f007:**
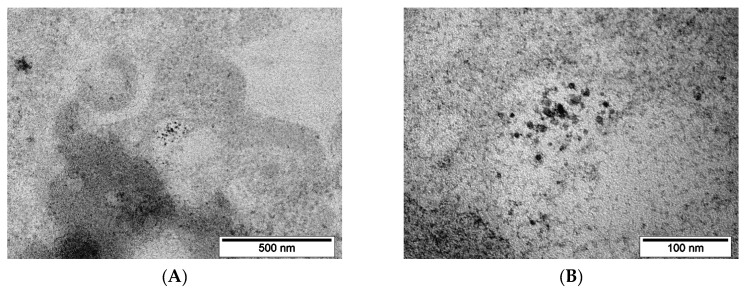
TEM images of the PC-3 cells after 2 h incubation with MNP-SS (**A**–**D**) or MNP-PEG (**E**–**H**): (**A**)—outer cell membrane has formed a vesicle around a group of MNP-SS nanoparticles; the group is enlarged in (**B**,**C**)—outer membrane and a cytoplasm of the PC-3 cell with attached clumps of MNP-SS nanoparticles and formed vesicle filled with nanoparticles; the vesicle is enlarged in (**D**,**E**,**G**)—PC-3 cells cytoplasm and plasmalemma with MNP-PEG nanoparticles, correspondingly; (**F**,**H**)—enlarged parts of (**E**,**G**), correspondingly.

**Figure 8 pharmaceutics-15-00127-f008:**
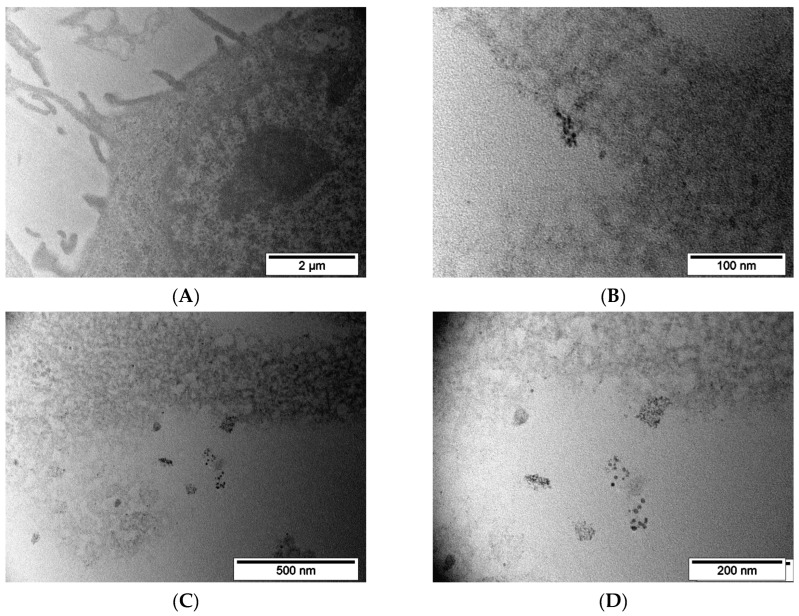
TEM images of the 4T1 cells after 2 h incubation with MNP-SS (**A**–**D**) and MNP-PEG (**E**–**H**): (**A**)—cytoplasm of 4T1 cell with a vesicle filled with MNP-SS nanoparticles; the vesicle is enlarged in (**B**,**C**)—outer membrane and cytoplasm of the 4T1 cell with attached clumps of MNP-SS nanoparticles and just formed vesicles; the vesicle is enlarged in (**D**)—filled with nanoparticles; (**E**)—4T1 cell plasmalemma with attached MNP-PEG nanoparticles, enlarged in (**F**–**H**)—MNP-PEG nanoparticles in the vicinity of the 4T1 cell plasmalemma.

**Figure 9 pharmaceutics-15-00127-f009:**
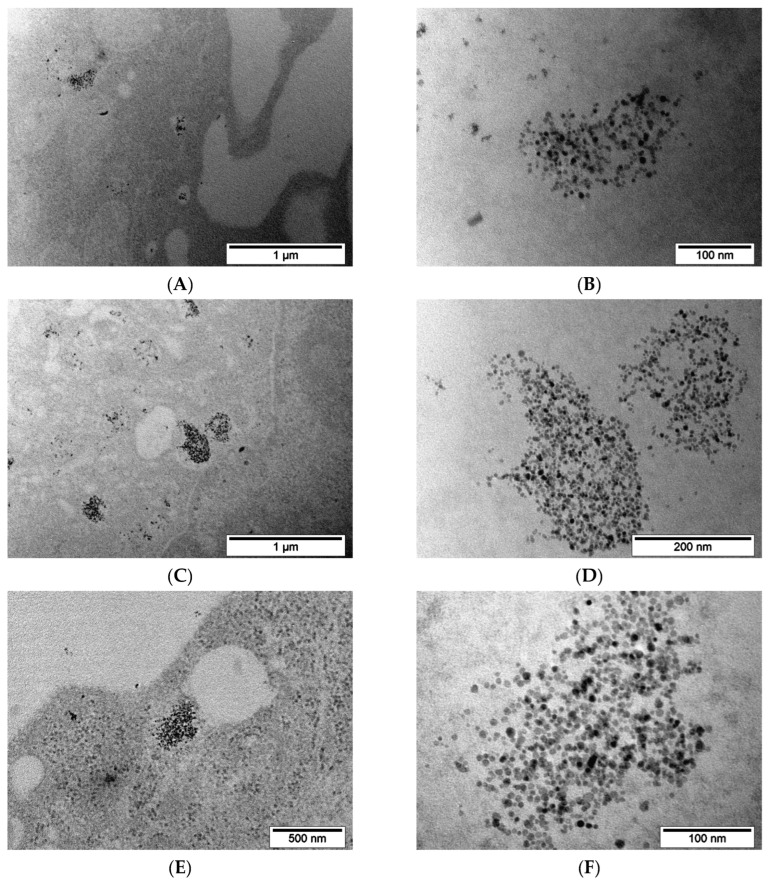
TEM images of the PC-3 cells after 24 h incubation with MNP-SS (**A**–**D**) or MNP-PEG: (**A**)—the outer cell membrane and a vesicle with numerous MNP-SS nanoparticles; the vesicle is enlarged in (**B**,**C**)—the low magnification overview of PC-3 cell with a lot of vesicles filled with MNP-SS nanoparticles; the vesicle is enlarged in (**D**,**E**,**G**)—PC-3 cells cytoplasm with a lot of vesicles filled with MNP-PEG nanoparticles; (**F**,**H**)—enlarged frames of (**E**,**G**), correspondingly.

## Data Availability

Data are available on request from the corresponding author.

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
