# Peer review of "Study of Cytotoxicity and Internalization of Redox-Responsive Iron Oxide Nanoparticles on PC-3 and 4T1 Cancer Cell Lines"

_pharmaceutics, 2022, doi:10.3390/pharmaceutics15010127_

Round 1
Reviewer 1 Report
It was a study about the evaluation of internalization and cytotoxicity of redox-responsive iron oxide nanoparticles against the PC-3 and 4T1 Cancer Cells. Here are some comments on this study that should be considered before publication:
1. Please prepare the paper in the format that the journal considered.
2. The abstract should be rewritten.
3. Please introduce all the abbreviations at their first-time usage.
4. “Despite the significant advantages of PEGylated compounds, the PEG coating adversely limits the rapid release of the drug from the delivery system due to its diffusion barrier, which can limit the use of PEG.” please add a reference related to this sentence.
5. There are some grammatical mistakes in the text that should be corrected.
6. For the material section it is better to separate them based on the manufacturer and write them as sentences.
7. Please add the manufacturer country of the instruments.
8. Please add TEM, VSM, and XRD results of nanoparticles after coating with PEG.
9. Please add statistical analysis for the MTS test. Moreover, 400 and 600 µg/ml are high, normally nanomaterials are used to decrease the concentration. How do you explain it?
Reviewer 2 Report
The manuscript reported as “Study of Cytotoxicity and Internalization of Redox-responsive Iron Oxide Nanoparticles on PC-3 and 4T1 Cancer Cell Lines” presents the synthesis and internalizations of magnetic nanoparticles into different cancer cell lines for cytotoxicity examination. The author utilized various surface-modulating agents on the magnetic nanoparticle and control the release of ions into the cellular structure. The paper in its current form is not publishable.
I recommend it for major revision. The author needs to clarify the following points.
1. The author investigates the effect of a surface-modulating agent during the intracellular uptaking process. however, only two surface ligands are used in this study. It is too narrow. Should consider surface-modifiers with various functional groups.
2. The current literature survey is not attractive. needs improvement.
3. Figure 1 needs to be revised. The author should consider the quality of the figure presentation.
4. The mean bar analysis and the number of experiments have to be mentioned at least in the figure caption.
5. Figure 3 contains a lot of information, however, the discussion is not sufficient. The author needs to provide a clear and concise explanation.
6. Why does the fluorescence intensity of Cyt 3 and Cyt 5 show the opposite versus incubation time?
7. Overall, the author needs to mention the novelty of this work at least in the abstract and conclusion. it is so unclear.
Reviewer 3 Report
The concept of introducing a disulfide bridge into a nanoparticle to target mainly cancer cells with toxic nucleus (iron nanoparticle to trigger ferroptosis) is of interest for potential cancer treatment with increased specificity. The authors provide all necessary details on the particle synthesis, properties, internalization into the cells. Cytotoxicity figure and some others, as well as materials and methods, need statistics approach description. English language needs proof-reading - preposition and articles are missing, etc.
Reviewer 4 Report
The manuscript is well written and the internalization dynamics of the MNP-SS and the cytotoxicity toward PC-3 and 4T1 cell lines were investigated in a proper way. The TEM analysis confirmed the absorption of cancer cells to the nanoparticles. Moreover, In Figure 7, 8 and 9, the high-resolution images and SAED patterns can add to verify the crystalline nature of the nanoparticles after absorbing cancer cells. The manuscript can be accepted after a minor revision.
Round 2
Reviewer 1 Report
-
Author Response
Thank you for your respond. Unfortunately it is difficult to improve the article because we didn’t get any concrete comments and/or suggestions.
Reviewer 2 Report
Thanks for providing your response to my concerns. I appreciate the list of changes. However, there are still some issues that need to improve. After lifting the following comments, it can be published.
1. Page 14 of the main manuscript is having a problem. The author needs to revise this section.
2. I am observing inappropriate pages bordering. The author should be careful while preparing the response letter.
3. I am not satisfied with Figure 1. I left the decision to the editor.
Author Response
Thank you for your comments and suggestions. Here are our responses.
Point 1
“1. Page 14 of the main manuscript is having a problem. The author needs to revise this section.”
Response 1
We are so sorry for this mistake. We’ve fixed it now.
Point 2
“2. I am observing inappropriate pages bordering. The author should be careful while preparing the response letter.”
Response 2
The main editor has said that all formatting issues will get fixed after the acceptance of the article by the editor’s team.
Point 3
“3. I am not satisfied with Figure 1. I left the decision to the editor.”
Response 3
We are sorry for that. We’ve tried as much as possible. We increased the quality of the Figure 1. We don’t know what else to change or improve, because the current remark is a bit abstract.